# Susceptibility of Female Mice to the Dietary Omega-3/Omega-6 Fatty-Acid Ratio: Effects on Adult Hippocampal Neurogenesis and Glia

**DOI:** 10.3390/ijms23063399

**Published:** 2022-03-21

**Authors:** Noelia Rodríguez-Iglesias, Agnes Nadjar, Amanda Sierra, Jorge Valero

**Affiliations:** 1Achucarro Basque Center for Neuroscience, Science Park of the UPV/EHU, E-48940 Leioa, Spain; noelia.rodriguez@achucarro.org (N.R.-I.); a.sierra@ikerbasque.org (A.S.); 2Department of Neuroscience, University of the Basque Country UPV/EHU, E-48940 Leioa, Spain; 3Neurocentre Magendie, U1215, INSERM-Université de Bordeaux, 33077 Bordeaux, France; agnes.nadjar@u-bordeaux.fr; 4Institut Universitaire de France (IUF), 33077 Bordeaux, France; 5Ikerbasque Basque Foundation for Science, E-48009 Bilbao, Spain; 6Institute of Neuroscience of Castilla y León (INCyL), University of Salamanca, E-37007 Salamanca, Spain; 7Institute for Biomedical Research of Salamanca (IBSAL), E-37007 Salamanca, Spain

**Keywords:** adult neurogenesis, dentate gyrus, diet, microglia, bacterial endotoxin lipopolysaccharide, LPS, omega 3, polyunsaturated fatty acids, sexual dimorphism, systemic inflammation

## Abstract

Maternal intake of omega-3 (n-3 PUFAs) and omega-6 (n-6 PUFAs) polyunsaturated fatty acids impacts hippocampal neurogenesis during development, an effect that may extend to adulthood by altering adult hippocampal neurogenesis (AHN). The n-3 PUFAs and n-6 PUFAs are precursors of inflammatory regulators that potentially affect AHN and glia. Additionally, n-3 PUFA dietary supplementation may present a sexually dimorphic action in the brain. Therefore, we postulated that dietary n-6/n-3 PUFA balance shapes the adult DG in a sex-dependent manner influencing AHN and glia. We test our hypothesis by feeding adult female and male mice with n-3 PUFA balanced or deficient diets. To analyze the immunomodulatory potential of the diets, we injected mice with the bacterial endotoxin lipopolysaccharide (LPS). LPS reduced neuroblast number, and its effect was exacerbated by the n-3 PUFA-deficient diet. The n-3 PUFA-deficient diet reduced the DG volume, AHN, microglia number, and surveilled volume. The diet effect on most mature neuroblasts was exclusively significant in female mice. Colocalization and multivariate analysis revealed an association between microglia and AHN, as well as the sexual dimorphic effect of diet. Our study reveals that female mice are more susceptible than males to the effect of dietary n-6/n-3 PUFA ratio on AHN and microglia.

## 1. Introduction

Omega-3 (n-3 PUFA) and omega-6 (n-6 PUFA) polyunsaturated essential fatty acids influence neurodevelopment [1,2,3]. Rodents and humans can only obtain n-3 PUFA and n-6 PUFA precursors from dietary sources; thus, during pregnancy, the n-3 PUFA and n-6 PUFA dietary content is a key determinant for brain formation [1,2,3]. However, Western diets are highly deficient in n-3 PUFA levels and are characterized by an unbalanced n-6/n-3 PUFA ratio that may compromise the adequate establishment of the brain cytoarchitecture and function [4,5]. The n-6/n-3 PUFA balance during prenatal development affects the hippocampus by modulating neurogenesis [2,6]. Relevantly, neurogenesis continues in the adult dentate gyrus of the hippocampus (DG) with the constant addition of new cells to the hippocampal circuitry, a process also known as adult hippocampal neurogenesis (AHN) [7]. Therefore, dietary n-6/n-3 PUFA balance during adult life might also affect the adult hippocampus structure, specifically in the DG where AHN occurs. Indeed, dietary n-3 PUFA supplementation increases AHN in rodents [8,9]. Additionally, the dietary n-6/n-3 PUFA ratio may also influence the brain through its immunomodulatory potential [10,11,12], as inflammation may compromise the plasticity and homeostasis of the central nervous system, e.g., affecting AHN, microglia, and astrocytes [13,14]. Moreover, microglia modulate AHN [15] and, therefore, inflammation may also affect AHN through its effect in microglia [13]. The n-3 PUFA-derived molecules may contribute to maintaining or even increasing AHN through their anti-inflammatory potential as they participate in the resolution of the inflammatory response. On the other hand, n-6 PUFA derivates mainly promote inflammation and, thus, may indirectly decrease neurogenesis [16,17,18]. Lastly, some reports suggest that n-3 PUFAs exert sexual dimorphic neuroprotective actions in the rodent central nervous system with a more significant impact on females than on males [19,20]. Thus, we postulated that dietary n-6/n-3 PUFA balance during adulthood might shape the DG in a sex-dependent manner by influencing AHN and cells involved in the brain’s inflammatory response (astrocytes and microglia).

To evaluate the role of dietary n-6/n-3 PUFA balance on AHN, microglia, and astrocytes, and the possible influence of sex, we fed female and male mice with a balanced (low n-6/n-3 PUFA ratio: 6.7) or n-3 PUFA-deficient (high n-6/n-3 PUFA ratio: >500) diet. We tested diet effects on neuroprogenitors, neuroblasts, and DG glial cells (microglia and astrocytes) in regular conditions and after a systemic and acute inflammatory event (bacterial endotoxin lipopolysaccharide, LPS, intraperitoneal administration) known to reduce AHN [21]. We observed that the n-3 PUFA-deficient diet reduced AHN most robustly in female mice, and it decreased microglia number and surveilled volume but did not affect astrocytes. Surprisingly, LPS administration triggered minimal effects, reducing the number of less mature neuroblasts. Lastly, our data suggest that dietary n-6/n-3 PUFA balance regulated the spatial interaction between microglia and neuroblasts, which may be relevant for neuronal maturation and survival [7]. Therefore, we show here that adult dietary n-6/n-3 PUFA balance strongly impacts the DG structure, affecting AHN and microglia, and that female mice are more susceptible than males to these effects.

## 2. Results

### 2.1. Dietary n-6/n-3 PUFA Content Did Not Affect LPS-Induced Weight Alterations

To test the effect of diet, LPS, and sex on the cellular composition of the adult DG, we fed female and male mice with balanced or n-3 PUFA-deficient diets from 6 weeks of age. Four weeks after the initiation of the diet, we injected mice with either saline or LPS intraperitoneally to induce an acute systemic inflammatory process known to affect the hippocampus [21] (Figure 1a).

We first monitored the well-known weight changes induced by LPS, which may influence behavior and, thus, neurogenesis [22]. First, we analyzed daily weight changes and the effects of dietary n-6/n-3 PUFA balance before LPS administration. Diet and sex (no interaction detected) significantly affected the weight of mice (two-way ANOVA, Appendix A). Before LPS administration, daily body weight gain was higher in males than in females. In addition, the n-3 PUFA-deficient diet increased the daily weight gain of the animals compared to the balanced diet in males (Figure 1b). We then checked daily body weight changes at three key timepoints after LPS administration in which weight loss and recovery occur: early (administration day and the following 2 days), mid (2 and 5 days after LPS administration), and late (from 5 days after LPS administration and for at least 36 days). Immediately after LPS administration, mice lost weight abruptly. During the 2 days after LPS administration (early post-LPS), we observed a significant interaction between sex and LPS treatment, but differences due to diet disappeared (three-way ANOVA analysis, Appendix A). LPS induced-weight loss was significantly higher in males than in females (Figure 1c). The 4 days after the initial weight loss, LPS treated mice slowly recovered weight, and sex and diet did not show any effect (Figure 1d). Lastly, 6 days after LPS injection, weight stabilized in LPS-treated mice. At this time, we did not observe any difference due to diet but detected the influence of sex and LPS (three-way ANOVA and two-way ANOVA, respectively, Appendix A), without relevant effects in post hoc analysis (Figure 1e). In summary, n-3 PUFA-deficient diet increased male mice daily weight gain before LPS administration but did not significantly affect body weight after LPS treatment. Our data indicate that dietary n-6/n-3 PUFA balance does not affect one of the most noticeable short-term effects induced by LPS administration: bodyweight oscillations.

### 2.2. Long-Term Effects of Dietary n-6/n-3 PUFA Balance on DG Size

Once we observed that the dietary n-6/n-3 PUFA ratio did not protect mice from short-term LPS-induced weight loss, we evaluated the long-term impact of diet, LPS administration, and sex on the brain. We first analyzed changes in the volume of the DG layers. We estimated the subgranular zone (SGZ), granular cell layer (GCL), and inner molecular layer (ML) volumes and detected differences only in the GCL (Figure 2 and Appendix A). We detected significant differences in the GCL due to sex and a significant interaction between diet and LPS treatment (three-way ANOVA and posterior two-way ANOVA, respectively, Appendix A). The n-3 PUFA-deficient diet reduced the GCL volume (Figure 2b) in males and females. Saline-treated female mice fed with the n-3 PUFA-deficient diet showed a smaller GCL volume than females fed with the balanced diet; such differences were not significant in females treated with LPS. In males, GCL was smaller in mice fed with the n-3 PUFA-deficient diet and treated with LPS compared to mice fed with the balanced diet and treated with saline. Our data revealed that the n-3 PUFA-deficient diet reduced the size of the GCL and suggested the existence of a sexually dimorphic effect.

### 2.3. Effects of Dietary n-6/n-3 PUFA Balance in Adult Hippocampal Neurogenesis

The decrease in GCL volume may be explained by a sustained reduction in the production of new neurons [23,24]. To explore this possibility, we evaluated the effects of dietary n-6/n-3 PUFA balance and LPS on AHN in female and male mice. First, we analyzed hippocampal radial neuroprogenitors, which are characterized by the expression of type VI intermediate filament protein nestin (neuroepithelial stem-cell protein) and glial fibrillary acidic protein (GFAP), as well as a radial morphology [25] (Figure 3a and Appendix A). We found a general effect of the diet and interaction between LPS administration and sex (three-way ANOVA, Appendix A). Therefore, we further analyzed our data to interpret LPS × sex interaction but did not detect any significant general effect of LPS or sex (two-way ANOVA, Appendix A). Hence, we finally pooled all data, grouping them by diet, and confirmed that the n-3 PUFA-deficient diet significantly decreased the number of radial progenitors in the DG (Figure 3b).

The reduction in neural radial progenitors induced by the n-3 PUFA-deficient diet may compromise the production of young neurons. To further explore this possibility, we used immunofluorescence detection of doublecortin (DCX)-expressing neuroblasts, which show different morphologies depending on their differentiation stage [26]. We estimated the total number of neuroblasts at different maturation stages distinguishing early neuroblasts (AB), intermediate neuroblasts (CD), and young neurons (EF, Figure 4a). We first analyzed the effects of the three variables (diet, LPS, and sex) on the number of early neuroblasts (AB) and found a significant reduction induced by the n-3 PUFA-deficient diet and LPS (Figure 4b) but not sex (three-way ANOVA followed by two-way ANOVA and post hoc analysis, Appendix A). Then, we quantified the intermediate neuroblasts (CD), which were reduced by the n-3 PUFA-deficient diet (Figure 4c), but not affected by LPS treatment or sex (three-way ANOVA followed by two-way ANOVA and post hoc analysis, Appendix A). Lastly, we analyzed young neurons (EF) and observed that LPS treatment did not significantly affect them, while diet exerted a sex-dependent effect on these cells (three-way ANOVA, significant interaction between diet and sex, Appendix A). In mice fed with the n-3 PUFA balanced diet, females showed more EF cells than males, whereas the deficient diet reduced the number of EF neuroblasts in female mice to the levels found in males (Figure 4d; two-way ANOVA and post hoc analysis, Appendix A). A similar effect of diet and sex was found when analyzing the total population of neuroblasts (Figure 4e). Our results indicate that diet and LPS treatment affected early neuroblasts, and that LPS effect disappeared in most morphologically mature neuroblasts, modulated by dietary n-6/n-3 PUFA balance in female but not in male mice.

To deeply explore the effects of diet, LPS, and sex in EF cells, we identified a subpopulation of intermediate maturing 2 week old neuroblasts by intraperitoneally injecting 5-bromo-2′-deoxyuridine (BrdU) into mice 2 weeks before their sacrifice (Figure 1a, Figure 4f and Appendix A). Most BrdU/DCX double-stained cells were morphologically mature 2 weeks after BrdU administration (Figure 4g), as expected, considering previously described maturation timing [26,27,28]. In these 2 week old EF cells, we found a significant interaction of diet and LPS, but no effect of sex (three-way ANOVA, Appendix A). The combination of n-3 PUFA-deficient diet and LPS treatment reduced the number of 2 week old BrdU/EF-DCX cells (Figure 4h) compared to mice fed with a balanced diet and/or treated with saline. Hence, our results indicate that the combination of n-3 PUFA-deficient diet with the LPS treatment reduced the initial number of DCX neuroblasts, and that this reduction extended to 2 week old EF-DCX cells.

Our results showed the absence of sex effects in early neuroblasts (AB, CD, and 2 week old EF cells) but demonstrated the existence of sexual dimorphism in the effects of diet in most mature ones. These results suggest that the dynamics of maturation and/or survival of neuroblasts may differ between female and male mice fed with a balanced diet (Figure 4b–d). Thus, as an indirect way to further explore changes in the dynamics of the maturation/survival of DCX-positive cells, we analyzed the proportion of each type of neuroblast per animal and observed significant differences due to diet and sex (no interaction detected), but not to LPS treatment (three-way ANOVA, Appendix A). Female mice fed with the n-3 PUFA balanced diet showed a shift in their proportion of the younger (reduced) and more mature type of neuroblasts (increased) compared to males fed with the balanced diet (Figure 4i). These data demonstrate the existence of sexual dimorphism in AHN dynamics, and that these dynamics are affected by dietary n-6/n-3 PUFA ratio in mice.

Changes described in young neurons should have relevant consequences in the codification of information reaching the DG. The probability of codifying information through newborn neurons may depend on the number and the extension of their dendrites [29,30]. Hence, we analyzed the area occupied by the DCX staining in the GCL and ML to indirectly evaluate the maximum potential of DCX cells to receive and process the incoming signals. Additionally, we obtained data from the SGZ, which mainly reflects the space occupied by the soma of neuroblasts. We found significant effects of diet in the three layers analyzed and interactions between sex and diet in the GCL and ML, with no effects due to LPS treatment (three-way ANOVA, Appendix A). In all the layers analyzed (SGZ, GCL, and ML), the n-3 PUFA-deficient diet decreased the extension of DCX cells, specifically in female mice (Figure 5). These results agreed with the previously described sex-dependent effect of dietary n-6/n-3 PUFA ratio on the number of EF cells. As expected, females fed with a balanced diet showed higher DCX staining in the three layers analyzed (SGZ, GCL, and ML) when compared to males. Furthermore, DCX staining also decreased to male levels in females fed with the n-3 PUFA-deficient diet in the SGZ, GCL, and ML.

To summarize, our data indicate that dietary balance in n-6/n-3 PUFA influences AHN, from neuroprogenitors to immature neurons. We also observed a detrimental effect of LPS in less mature neuroblasts but not in the more mature ones. Importantly, dietary n-6/n-3 PUFA balance showed the most significant impact exclusively in the DG of female mice, where the balanced diet increased the total amount of young neurons and the extension of neuroblasts (compared with males fed with balanced or deficient diets and females fed a deficient diet). Thus, we demonstrated that female mice are more susceptible than males to the effects of dietary n-6/n-3 PUFA balance in AHN.

### 2.4. Effects of Dietary n-6/n-3 PUFA Balance on DG Microglia

Microglia exert a relevant role in surveilling the brain parenchyma and mounting the inflammatory response when necessary [31]. At the same time, microglia participate in the regulation of AHN [7,15]. Considering that dietary n-6/n-3 PUFA balance may modulate the inflammatory response [16,17,18] and, thus, affect microglia, we analyzed the effects of balanced and n-3 PUFA-deficient diets in combination with LPS in DG microglia number and extension.

To check the effects of diet, LPS treatment, and sex in microglia, we estimated their number in the SGZ, GCL, and ML. We detected significant effects of the diet in the GCL and ML (not in the SGZ), and no effects due to LPS or sex (three-way ANOVA, Appendix A). Therefore, we focused our analysis on the effects of the diet, pooling our data by diet, and we observed that the n-3 PUFA-deficient diet reduced the number of microglia in the GCL and ML but not in the SGZ (Figure 6a,b and Appendix A). Next, we indirectly evaluated microglial surveillance capacity by analyzing the percentage of area occupied by the GFP staining in the DG (surveilled volume). The diet exerted an effect in all the layers analyzed, sex showed an effect in the GCL and ML (not in the SGZ), and we did not detect significant effects due to LPS treatment (three-way ANOVA, Appendix A). Therefore, we focused on analyzing effects due to diet and sex. Microglia covered less space in both female and male mice fed with the n-3 PUFA-deficient diet in the three layers of the DG (SGZ, GCL, and ML, Figure 6c), indicating that their function may be compromised. Although sex effect was statistically significant (two-way ANOVA, Appendix A), no relevant differences between males and females were detected in the post hoc analysis, suggesting a weak sexual dimorphism. Summarizing, our results indicated that a dietary deficiency in n-3 PUFAs compromises microglia number and surveillance capacity in the DG. 

Microglia affect AHN, and their interaction with maturing neurons may be relevant during neuronal maturation [7,15,32]. As this interaction may depend on the surveillance capacity of microglia, we explored microglial surveillance of neuroblasts by analyzing colocalization between DCX (neuroblasts) and GFP (microglia) staining (Figure 7 and Appendix A). We observed a significant global effect of the diet in microglial surveillance of DCX cells in the SGZ, with no effects due to sex or LPS treatment (three-way ANOVA, Appendix A). Additionally, we found significant differences due to sex in the GCL, and no effects due to diet or LPS (three-way ANOVA, Appendix A). We did not detect significant differences in the ML. Therefore, we focused our analysis on diet and sex effects in the SGZ and GCL. We observed an almost significant reduction in neuroblast surveillance in the SGZ of male mice fed with n-3 PUFA-deficient diet (*p* = 0.0521) compared to male mice fed with the balanced diet, as well as a trend toward reduction in the GCL of male mice compared to females (*p* = 0.0594), both fed with the n-3 PUFA-deficient diet (Figure 7b). To better explore the relevance of these results, we analyzed the randomness of the observed colocalization by comparing actual colocalization data with data obtained after rotating microglial staining (see Section 4), exploring diet effects separately in females and males. Our analysis revealed that the proportion of DCX staining area colocalizing with microglia was higher than expected just by chance, indicating that these data may reflect active microglial surveillance of DCX cells. In addition, we observed differences due to diet in rotated and real data. These analyses confirmed that the n-3 PUFA-deficient diet reduced microglial surveillance of neuroblasts (Figure 7c).

Our results indicate that the imbalance in n-6/n-3 PUFA reduces the number of microglia in the DG, compromising their capacity to surveil the brain parenchyma and, specifically, regions occupied by neuroblasts.

### 2.5. Effects of Dietary n-6/n-3 PUFA Balance in Astrocytes

Astrocytes are the most abundant glial cells in the brain and may respond to inflammatory challenges [33]. Considering the role of n-6/n-3 PUFA balance in the regulation of AHN, which also contributes to the generation of astrocytes [34], and its role in modulating the inflammatory status [16,17,18], we analyzed the possible effect of our diet in the DG population of GFAP-positive astrocytes. We observed a general effect of LPS treatment in the SGZ, but no effects due to diet or sex (three-way ANOVA, Appendix A), indicating that LPS may reduce astrocyte number (no significant differences in subsequent *t*-test analysis were found; Figure 8). In the GCL, we detected an interaction between diet and sex (no significant differences in subsequent one-way ANOVA analysis, Appendix A) and no effect due to LPS treatment (three-way ANOVA, Appendix A). Our data suggest that the population of GFAP-expressing astrocytes is not relevantly affected by dietary n-6/n-3 PUFA balance.

### 2.6. Multivariate Analysis of the Data

Globally, we showed that dietary n-6/n-3 PUFA balance affects DG volume, AHN, and microglia, and we detected the interaction of diet with the two other independent variables analyzed: sex and LPS. However, classical statistics do not capture possible relationships between the different dependent variables studied. Thus, we performed a correlation matrix test to evaluate global relationships between dependent variables, i.e., features. Additionally, we performed a PCA and a subsequent clustering analysis to better understand the combined effect of diet, LPS treatment, and sex in mice and the relevance of the features evaluated. 

To facilitate the interpretation of the correlation matrix analysis, we grouped the parameters evaluated in five main categories: weight change, DG layer volume, adult neurogenesis, microglia, and astrocytes. Correlation matrix analysis revealed significant correlations between parameters of the same group: daily change in weight at early and intermediate times after LPS administration, the volumes of the different layers of the DG, the number of different types of neuroblasts and the space they occupied in the DG layers, the number of microglia and the area they surveilled (total and of DCX cells), and the number of astrocytes in the different layers of the DG (Figure 9). The proportion of significant intergroup correlations oscillated between 100% and 16% (weight change: 16.67%; DG layer volume: 100%; adult neurogenesis: 31.43%; microglia: 39.39%; astrocytes: 100%; percentages of significant correlations referred to the total number of possible correlation pairs in each group). Interestingly, cross-correlations between different categories were mainly observed between adult neurogenesis and microglia (25.56% of all possible pairs of correlations were significant, Appendix A) and between DG volumes and astrocytes (22.22% of all possible pairs of correlations were significant, Appendix A). Relevantly, the number of DCX cells and their extension positively correlated with microglia and the proportion of area they were surveilling (Figure 9). In contrast, astrocytes did not show such a strong correlation with neurogenesis, and just the number of most immature DCX neuroblasts (AB) and the size of the SGZ and ML positively correlated with the number of astrocytes (2.22% of all possible pairs). Concluding, our correlation analysis mainly reflected an association between microglia and neurogenesis parameters. 

To understand the relative weight of each variable (diet, LPS, and sex) on the variability of the data, we performed a PCA. We conducted the PCA with 16 features selected on the basis of their information gain ratio (see Section 4 and Appendix A), a proxy of how much information about the independent variables each feature contained. Principal components 1 and 2 (PC1 and PC2), which explained 62% of the data variance, allowed independent clustering of mice fed with balanced and n-3 PUFA-deficient diets (Figure 10a). Hence, PCA confirmed that n-6/n-3 PUFA dietary balance significantly impacts the DG of mice. Additionally, PC1 and PC2 showed a relevant influence of sex, as female mice fed with the balanced diet were better separated than males from mice of both sexes fed with the deficient diet (Figure 10b). To deeply analyze the contribution of diet and sex, we then used a PCA heat map to perform cluster analysis of all the mice combining the first three PCs. Two main clusters were obtained (Figure 10c). One of the clusters included mainly female mice fed with a balanced diet, reflecting the specific susceptibility of female mice to diet. The other cluster contained a mixture of female mice fed with a deficient diet and male mice fed with either balanced or n-3 PUFA-deficient diets. The clusters obtained included a combination of LPS-treated and untreated mice, suggesting relatively weak effects of acute systemic inflammation compared with the effects of diet and sex. PCA indicated that diet strongly influenced the 16 parameters evaluated, especially in female mice. 

We further analyzed the relevance of the features included in the PCA. Adult neurogenesis and microglial surveillance-related features highly influenced PC1 (which explained 42% of the data variability). PC2 (which explained 20% of the variability) was mainly affected by the volume of the SGZ, features related to the intermediate neuroblasts (CD), and the DCX area surveilled by microglia (Figure 10c). Thus, neurogenesis and microglial-related features were responsible for most of the observed variability.

Our multivariate analysis revealed important relations between microglia and neuroblasts and significant effects of dietary n-6/n-3 PUFA balance, especially in female mice. Altogether, our results demonstrated that the n-3 PUFA-deficient diet negatively impacted the cellular composition of the adult DG, downregulating AHN and impairing microglia surveillance, and that it had a more significant impact in female mice.

## 3. Discussion

We demonstrated that dietary n-6/n-3 PUFA balance modifies the cytoarchitecture of the adult DG affecting the size of the GCL, AHN, and microglia. We observed weak effects due to LPS administration and modulatory effects of dietary n-6/n-3 PUFA balance. Our most remarkable result is that the impact of dietary n-6/n-3 PUFA balance in AHN was sexually dimorphic. Importantly, our multivariate analysis indicated that female mice showed more susceptibility than males to changes induced by diet.

We previously described that a single LPS administration, similar to that used here, had a long-term negative effect on AHN [21]. In agreement, in the current study, LPS decreased neuroblasts (AB cells) and a subpopulation of 2 week old immature neurons (EF BrdU/DCX positive cells, Figure 4). However, we did not detect any effect of LPS in the total amount of young neurons (EF DCX cells), as we did in our previous study. Differences between the batches of LPS used may account for the divergence in the strength of the long-term effects of LPS administration between studies. Additionally, in our previous study, our mice were housed under mild environmental enrichment conditions, which may have potentiated adult neurogenesis and made the long-term effects of LPS administration more obvious. Another plausible interpretation of our current results is that the strong effects of the diet in AHN may have masked the reduction induced by LPS in the most mature type of DCX cells (EF cells) [21]. Indeed, the n-3 PUFA balanced diet abolished the LPS-induced reduction in early neuroblasts and 2 week old EF cells (Figure 4b,h). In agreement, others have previously demonstrated in vitro and ex vivo that n-3 PUFA derivates prevent the detrimental effects of LPS in neurogenesis [35,36,37]. As systemic inflammation induced by LPS may also influence glia, we checked astrocytes and microglia in the DG. Our results about diet and LPS treatment effects in astrocytes were not conclusive as we only detected weak and inconsistent effects of LPS treatment in the SGZ and diet in the GCL. We did not detect any long-term effect of LPS treatment in microglia, although we described an increase in the proportion of newly generated microglia in our previous study [21]. Again, this lack of detection of long-term effects of LPS may be related to differences in LPS batches, mouse housing conditions, and/or the possible masking action of the effects exerted by the diet in microglia. As evidenced by the weak influence of LPS administration in our PCA, further studies will be required to adequately dissect the modulatory action of dietary n-6/n-3 PUFA balance on the long-term effects of LPS in the DG. 

We described that adult dietary n-6/n-3 PUFA balance impacts the cellular composition of the DG, affecting neuroblasts and microglia. First, we showed that the dietary imbalance in n-6/n-3 PUFA decreased the number of hippocampal neuroprogenitors. In agreement, docosahexaenoic acid (DHA), one of the derivates of n-3 PUFA, increases neuroprogenitor proliferation at low concentrations and promotes differentiation at higher ones [38]. Indeed, n-3 PUFA derivates promote neurogenesis in vitro without changing gliogenesis [39,40,41,42], which matches our observations in vivo (Figure 8). The impact of the diet on neuroprogenitors extended to neuroblasts, as we also showed that the n-3 PUFA-deficient diet reduced the number of early neuroblasts (Figure 4). Others previously found an increase in DCX-positive neuroblasts in adult mice when fed with diets with a lower n-6/n-3 PUFA ratio (0.5) than the one used here (6.7) [43]. In general terms, evidence suggests that decreasing dietary n-6/n-3 PUFA ratio increases the number of newly generated neurons in the DG and vice versa (increasing dietary n-6/n-3 PUFA ratio decreases AHN). The molecular mechanisms governing the effects of PUFA in adult neurogenesis are not fully understood. Here, we discuss those mechanisms involved in stem-cell-cycle regulation, neural differentiation, and cell survival. Both n-3 PUFA derivates, DHA and eicosapentaenoic acid (EPA), induce cell-cycle arrest in cultured embryonic stem cells by reducing the expression of the hairy and enhancer of split 1 (Hes1) transcription factor [41]. DHA upregulates the canonical WNT/β-catenin pathway, a well-known regulator of adult neurogenesis [44], in neural progenitors derived from human-induced pluripotent stem cells [45]. Furthermore, DHA increases CREB (cAMP response element-binding protein) activity in cell cultures of neuroprogenitor cells [42,45], and CREB promotes the differentiation and survival of newborn neurons [46]. Additionally, PUFA may modulate neurotransmission and synapse formation in maturing neuroblasts. DHA and the n-6 PUFA derivate arachidonic acid directly modulate the activity of ion channels such as the voltage-gated potassium channel, thus influencing neurotransmission [47,48]. Moreover, an n-3 PUFA deficient diet reduces synaptosome levels of hippocampal glutamate receptors (GluA1, GluA2, and NR2B) [49], which are involved in synaptic plasticity and neuronal maturation during adult neurogenesis [7]. Microglia may also be influenced by n-PUFA intake, as they accumulate n-PUFA derivatives in their membrane [50]. The n-3 PUFA-deficient diet significantly reduced microglia in the GCL and ML (Figure 6). Among other plausible explanations, this effect of the n-3 PUFA-deficient diet may be related to lipoxin A4 (LxA4), one of the derivates of n-6 PUFA with anti-inflammatory action, that may be elevated in mice fed with the deficient diet [51]. LxA4 reduces microglia in the context of neurodegeneration [52,53,54] and may have exerted a similar effect in our mice. We do not have data about whether LxA4 levels were changed in our mice after feeding them with the n-3 PUFA-deficient diet; thus, we leave this hypothesis open to be explored in future work. Other studies have shown a decrease in the density of microglia when supplementing mice diet with n-3 PUFA, as reviewed in [55], contradicting our results. This discrepancy may be attributed to the different experimental settings used between others and our present study, which differ in the initiation of n-6/n-3 PUFA ratio-controlled diets. As we focused on the effects in adulthood, our diet was initiated 3 weeks after weaning, while most of the other studies focused on the effects of maternal diets on the offspring. Moreover, we also showed that the n-3 PUFA-deficient diet reduced the area occupied by microglia, thus affecting microglia surveillance, which agrees with previous results indicating that maternal n-3 PUFA-deficient diets decrease microglial motility in the postnatal brain [56]. Lastly, we indicated that dietary n-6/n-3 PUFA balance influenced the interaction between microglia and neuroblasts based on our colocalization analysis. Furthermore, matrix correlation analysis showed consistent correlations between microglia and neurogenesis-related parameters, reinforcing the idea of the communication between microglia and neuroblasts. A previous study revealed that maternal dietary n-6/n-3 PUFA balance regulates microglial interaction with maturing neurons during hippocampal development by acting in microglial phagocytosis of synaptic regions, affecting neuronal morphology and hippocampal function, an effect partly mediated by the oxylipin 12/15-lipoxygenase/12-HETE signaling pathway [2]. Although further analyses are required to confirm if similar effects occur due to differences in dietary n-6/n-3 PUFA balance during adulthood, these data suggest that changes detected here in microglial surveillance may interfere with the final steps of young neurons maturation (regulation of young neurons connectivity), altering neuronal survival and, thus, AHN [7]. In summary, all these data indicate that dietary n-6/n-3 PUFA balance may modulate adult neurogenesis by regulating the interaction between microglia and neuroblasts. 

We demonstrated relevant sex differences in the effects of dietary n-6/n-3 PUFA balance in adult neurogenesis. We observed differential effects of diets between female and male mice in the most morphologically mature population of DCX neuroblasts and the area occupied by neuroblasts (Figure 4d and Figure 5). Previous studies have shown that neuroprotective effects of dietary supplementation with n-3 PUFA or its derivate DHA differ in females and males [19,20]. Relevantly, sex hormones such as progesterone may interact with DHA levels by increasing its synthesis such that females show higher conversion of n-3 PUFA into DHA [57]. This action of sex hormones may explain our results, as the effect of n-3 PUFA balanced diet should be most potent in females compared to male mice due to a higher conversion of n-3 PUFA into its proneurogenic derivate DHA. The existence of sexual dimorphism in adult neurogenesis is controversial as some reports indicated no differences in rodents, while others showed sex differences dependent on the phase of the estrous cycle, as reviewed in [58]. Importantly, we only detected AHN sexual dimorphism in mice fed with the balanced diet but not in mice fed with the n-3 PUFA-deficient diet, without considering the estrous cycle. Thus, variations in diet may explain some of the contradictory results obtained about sexual dimorphism in mouse adult neurogenesis. For example, a previous study showed that the total number of DCX neuroblasts does not differ between female and male mice [59], which aligns with our data in mice fed with the deficient diet, but does not coincide with our data from mice fed with the balanced one. Nevertheless, as the composition of the diet used to feed mice is not normally disclosed in the publications, we cannot evaluate if the lack of detection of sexually dimorphic effects in adult neurogenesis was related to a dietary deficiency in n-3 PUFA. Hence, our results highlight the importance of considering the effects of the dietary content in n-6 PUFA and n-3 PUFA when exploring sexual dimorphism. Most relevantly, our study reveals that the hippocampus of female mice is more susceptible to changes in dietary n-6/n-3 PUFA balance than that of male mice.

## 4. Materials and Methods

### 4.1. Mice

All experiments were performed in female and male mice in which microglia expressed the green fluorescent protein (GFP; fms-EGFP or MacGreen mice) [60]. Mice were kept under standard housing conditions with a 12 h light/dark cycle and ad libitum access to food and water. All procedures followed the European Directive 2010/63/EU and were approved by the Ethics Committees of the University of the Basque Country EHU/UPV (Leioa, Spain; CEEA M20/2019/279 and CEIAB M30/2019/276).

### 4.2. Diets

After weaning (at 3 weeks of age) and until 6 weeks of age, female and male mice were fed with regular chow (Teklad Global 14% Protein Rodent Maintenance Diet, Teklad diets, Envigo) and then fed diets with either a low (n-3 PUFA balanced) or high (n-3 PUFA-deficient) n-6/n-3 PUFA ratio (INRAE, Jouy-en-Josas, France) for 10 weeks (Figure 1a). The balanced and n-3 PUFA-deficient diets were isocaloric, containing 5% fat, and they differed in their linoleic acid/α linoleic acid (LA/ALA) ratio (Table 1 and Table 2). The diet of breeding mice presented an n-6/n-3 PUFA ratio of 10.3 (Teklad Global 18% Protein Rodent Diet, Teklad diets, Envigo), the regular diet ratio was 20, the n-6/n-3 PUFA ratio of the deficient diet was over 500, and the balanced diet ratio was 6.7.

### 4.3. LPS and BrdU Administration

Four weeks after the beginning of the diets, mice received a single intraperitoneal injection of LPS (1 mg/kg; L6143, Sigma-Aldrich, Madrid, Spain) to induce acute systemic inflammation (Figure 1a). Control animals were injected with saline (phosphate-buffered saline, PBS; pH 7.4). We monitored and weighed mice daily before and after the LPS injection and estimated net daily changes in weight for four different periods: (1) pre-LPS (mean daily weight changes monitored for, at least, 25 days before LPS administration), (2) early post-LPS (mean weight change per day between LPS administration day and the following 2 days), (3) mid post-LPS (daily weight change 2 and 5 days after LPS administration), and (4) late post-LPS (daily weight change monitored 5 days after LPS administration and for, at least, 36 days).

To analyze the progression of neuronal maturation, we tagged proliferating cells 2 weeks before mice sacrifice (Figure 1a). We used an accumulative administration protocol of BrdU (B5002-5G, Sigma-Aldrich) consisting of four intraperitoneal injections of BrdU (100 mg/kg, diluted in 0.1% NaOH, PBS) every 2 h.

### 4.4. Transcardial Perfusion and Brain Tissue Collection

Four month old mice were anesthetized with a single injection of avertin (2,2,2-tribromoethanol, Sigma-Aldrich) and transcardially perfused with PBS followed by 4% paraformaldehyde in PBS (4% PFA). Brains were collected and post-fixed in 4% PFA for 3 h. We cut left hemispheres in 50 µm thick sagittal sections using a Leica VT 1200S vibrating blade microtome (Leica Microsystem GmbH, Wetzlar, Germany). 

### 4.5. Immunofluorescence

Vibratome sections were rinsed (3×) in PBS, incubated in 2 N hydrochloric acid for 30 min at 37 °C, rinsed (3×) with 0.1 M borate buffer pH 8.5 and PBS (3×), and then blocked in permeabilizing/blocking solution (0.2% Triton X-100, 3% bovine serum albumin in PBS; Sigma-Aldrich) for 1 h at room temperature. Two different series of sections of each animal were incubated for 2 days at 4 °C in constant shaking with one of the following combinations of antibodies diluted in the permeabilizing/blocking solution: (a) anti-BrdU (1:1000 made in rat, NB500-169, Novus Bio, Abingdon, United Kingdom) to detect 2 week old cells, anti-DCX (1:1000 made in goat, SC-8066, Santa Cruz Biotechnologies, Heidelberg, Germany) to detect maturing neuroblasts, and anti-GFP (1:1000 made in chicken, GFP-1020, Aves Labs, Davis, CA, USA) to detect microglia; (b) anti-BrdU, anti-GFAP (1:1000 made in rabbit, DAKO 20334, Agilent Technologies, Santa Clara, CA, USA) to detect astrocytes and radial neuroprogenitors, and anti-nestin (1:1000 made in chicken, NES, Aves Labs) to detect radial neuroprogenitors. After incubation with primary antibodies, sections were rinsed (3×) with PBS and incubated for 2 h at room temperature with 4,6′-diaminide-2-phenylindole (DAPI, 5 mg/mL, D9542, Sigma-Aldrich) and the corresponding secondary antibodies (1:500, conjugated with Alexa-488, Rhodamine red or Alexa-647, Jackson ImmunoResearch, Cambridgeshire, United Kingdom) diluted in the permeabilizing/blocking solution. After washing (3×) with PBS, we mounted sections on glass slides with DAKO mounting medium (Agilent Technologies). 

### 4.6. Image Collection

All images were obtained using a laser-scanning microscope (Leica SP8, 40× oil-immersion objective). Five images of the DG (one of the horn and two from each arm) were acquired from the three central sections of the septal DG (*z*-stack = 12 slices, *z*-step = 1.15 µm, *xy* resolution: 512 × 512 pixels and 1.76 µm/pixel). We also obtained images using an epifluorescence microscope (Zeiss Axio Observer.Z1, 10× objective, *xy* resolution: 1.55 µm/pixel, Carl Zeiss Microscopy Deutschland GmbH, Oberkochen Germany) to estimate septal DG volumes. All images were imported into Fiji software [61] in tiff format as 8 bit images for further quantification. 

### 4.7. Cell Quantification

Cells were manually quantified using Fiji software and following optical dissector stereology guidelines. Cells located in exclusion zones (top confocal plane and left and bottom *xy* borders of the image) were not quantified to avoid overestimation errors. Septal DG cell numbers were estimated using the following equations:Total cells=Volumetric density × Volume,
$$\mathrm{Volumetric}\;\mathrm{density}=\frac{\Sigma \;\mathrm{cells}}{\Sigma \mathrm{Areas}\;\times \;\mathrm{Thickness}},\;$$
where *Areas* are the surfaces of the DG regions (SGZ, GCL, or ML) in which we quantified the cells, and *Thickness* is the sum of every *z*-step in which cells were quantified (11 from a total 12) in each stack of images (12.65 µm per stack). The areas of the analyzed regions were measured by creating regions of interest (ROIs) in the middle slice of the stack of images. For the ML, we selected a 40 µm thick band from the outer limit of the GCL. For the SGZ, we delineated a 16 µm thick band around the inner limit of the GCL (Figure 5a).

Volumes were estimated using the following Equation:$$\mathrm{Volume}=\frac{\Sigma {\mathrm{Areas}}^{\prime }\times {\mathrm{Thickness}}^{\prime }}{\;\mathrm{Quantified}\;\mathrm{fraction}},$$
where *Areas*^′^ is the total surface of the DG regions analyzed in each section, *Thickness*^′^ is the section thickness (50 µm) immediately after cutting, and *Quantified fraction* is the proportion of quantified sections (one out of six).

Septal DG DCX-positive cells were classified into three categories on the basis of their morphology [26]: AB (bipolar cells with processes parallel to the central axis of the GCL), CD (bipolar cells extending a process into the GCL), and EF (ramified cells extending their dendrites into the ML). The volumetric density of cells and the total number were estimated using GCL volumes. Additionally, the total amount of BrdU-positive cells was similarly estimated, and the proportion of BrdU cells showing DCX staining was calculated to finally extrapolate the total amount of double-stained BrdU/DCX cells of each category. Radial neuroprogenitors were identified as GFAP-positive cells with their soma in the SGZ and an apical nestin-positive process extending through the GCL. Microglia (GFP-positive cells) and astrocytes (GFAP-positive cells with a stellate morphology) were also manually quantified in the SGZ, GCL, and ML. The total number of radial neuroprogenitors, microglia, and astrocytes was estimated following the same steps used for DCX-positive cells. Colocalization for BrdU was also analyzed for these cells, but the number of colocalizing cells was not enough to extract valid conclusions about changes in the generation of microglia or astrocytes.

### 4.8. Analysis of the Extension of Doublecortin Staining, Microglia Surveilled Volume, and Colocalization

We followed a semi-manual detection protocol to analyze DCX (neuroblasts) and GFP (microglia) staining. We first pretreated the images to reduce background and noise using a difference of Gaussians (DoG) filter (minimum sigma 1 for DCX detection and 0.5 for GFP, maximum sigma 30 for DCX and 10 for GFP; values were adjusted empirically visualizing their effects on the images). Then, a manual threshold was established (around 5 for DCX staining and 3 for GFP). This threshold was used to generate a binary mask with the analyze particles tool from Fiji, removing regions smaller than 1.5 µm^2^ for DCX and 1 µm^2^ for GFP. Then, we filtered the mask using a three-dimensional Close filter to soften the edges of the masked areas. The Close filtering consisted of sequentially applying maximum and minimum filters with a radius of 1 µm, except for the *x-* and *y*-values of the minimum filter for DCX, which were 1.57 µm. Then, one ROI per image slice was generated from the filtered mask, and the matching between the ROIs and the original images staining was visually checked. When ROIs poorly corresponded with the staining, the process was repeated, and the threshold selection was changed until a proper match between ROIs and staining was obtained. Finally, ROIs were saved for further analysis.

The ROIs of DCX and GFP staining were combined with the ROIs of the different layers analyzed (SGZ, GCL, and ML) using the ROI Manager function “AND” to generate ROIs of the staining of each particular region. The areas of the ROIs of layers and staining were measured, and the percentage of area occupied by the staining was calculated for the three different layers analyzed. For colocalization analysis, ROIs corresponding to DCX and GFP staining were combined using the ROI Manager function “AND” to obtain the ROIs of the colocalizing areas. Then, we analyzed each DG region separately by combining the whole image ROIs of the colocalizing areas, DCX staining, or GFP staining with the ROIs of each layer using the “AND” function. Finally, the percentage of DCX and GFP areas occupied by the colocalizing ROIs was calculated with the following equations:$$\mathrm{DCX}\;\mathrm{area}\;\mathrm{surveilled}\;\mathrm{by}\;\mathrm{microglia}=\frac{\Sigma \mathrm{Colocalizing}\;\mathrm{ROI}\;\mathrm{areas}\;}{\;\Sigma \mathrm{DCX}\;\mathrm{ROI}\;\mathrm{areas}}\times 100,$$
$$\mathrm{Microglia}\;\mathrm{surveilling}\;\mathrm{DCX}\;=\frac{\Sigma \mathrm{Colocalizing}\;\mathrm{ROI}\;\mathrm{areas}\;}{\;\Sigma \mathrm{GFP}\;\mathrm{ROI}\;\mathrm{areas}}\times 100.$$

To evaluate if the obtained colocalization was the one expected by chance, we split GFP ROIs using the Split function of ROI Manager and rotated them around their center in steps of 5° from −90° to 90°. Finally, we calculated the percentage of colocalization per image obtained after each rotation step. 

### 4.9. Statistical Analysis

All analyses were performed using Graphpad Prism 9 software (San Diego, CA, USA) except for the PCA, which was carried out using Orange 3.29 software [62].

Considering the complexity of statistically analyzing the effects of the combination of three independent variables (i.e., diet, LPS treatment, and sex), we developed a stepwise analysis protocol (all data and analysis available in Appendix A). First, a three-way ANOVA analysis was performed to detect general effects and possible interactions among the three independent variables analyzed. If, at this point, a significant interaction between two of the three parameters was detected, a two-way ANOVA was performed to interpret significance adequately. Usually, one of the factors (sex or LPS treatment) did not show a significant effect; thus, data were grouped (consolidated), and a two-way ANOVA was performed to analyze general effects and interaction between the other two variables. A one-way ANOVA was performed to interpret significance adequately if a significant interaction between parameters was detected in the two-way ANOVA. Finally, Tukey’s post hoc analysis was performed. The normality of the residuals was analyzed using the D’Agostino–Pearson omnibus (K^2^) test, and Spearman’s test was used to evaluate heteroscedasticity. If normality of the residuals or heteroscedasticity tests failed, data were transformed as indicated in the Appendix A to comply with these criteria. In the case of one-way ANOVA analysis, equality of standard deviations (SDs) was tested using the Brown–Forsythe test, and data were transformed when needed to reach this criterion as indicated for each particular case in the Appendix A. When only diet effects were significant, data from different sexes and from saline- and LPS-treated mice were consolidated and used to perform a two-tailed unpaired *t*-test. In these particular cases, the F-test was used to analyze the equality of the variances, and the estimation plot was used to visualize significant differences by plotting the 95% confidence intervals (CIs) of the difference between means. Differences between groups were only considered to be statistically significant when *p* < 0.05. Data are shown as the mean ± SEM, and each datum is represented in superimposed dot plots when possible. 

Colocalization randomness was analyzed by plotting cumulative frequencies of real vs. rotated data and their 95% CIs. The departure of randomness was considered when a consistent region of the real data frequencies was not included between the CIs of the corresponding rotated data. 

Correlation matrix analyses were performed using the nonparametric Spearman’s *r* test to avoid the confounding effect of clustered data. Correlations were considered significant only when *p* < 0.01 to reduce random detection of unfair significance due to multiple comparisons. 

For PCA, features were normalized using standardization to mean = 0 and SD = 0. Then, the 16 features with the best gain ratios [63], out of 34, were selected as those carrying the most relevant information (we excluded weight change features after LPS administration as they have been previously described as hallmarks of the LPS effect); selected features and their gain ratios are listed in Appendix A. These features were checked to ensure they included at least one feature from the following categories: weight differences, neurogenesis, volume changes, microglia, and astrocytes. Then, a PCA was performed with the selected 16 features, and the first two components were analyzed and plotted. We used components and transformed data to visualize the contribution of features to each component and mice clustering, respectively. Finally, the three first components were analyzed using heat map clustering of mice.

## 5. Conclusions

Our results indicate that dietary n-6/n-3 PUFA balance during adulthood significantly impacts the production of new neurons in the DG of the hippocampus, and that diet also affects microglia surveillance potential. We also detected some modulatory effects of n-6/n-3 PUFA balance in the decrease induced by LPS in AHN, although this needs further exploration. Notably, we demonstrated that the effects induced by dietary n-6/n-3 PUFA balance are more prominent in females than in male mice. This study reveals the importance of dietary n-6/n-3 PUFA balance for the structural and likely functional state of the DG. Therefore, we propose that an adequate dietary n-6/n-3 PUFA balance in females may contribute to maintaining brain plasticity and homeostasis through adult neurogenesis and microglia regulation.

## Figures and Tables

**Figure 1 ijms-23-03399-f001:**
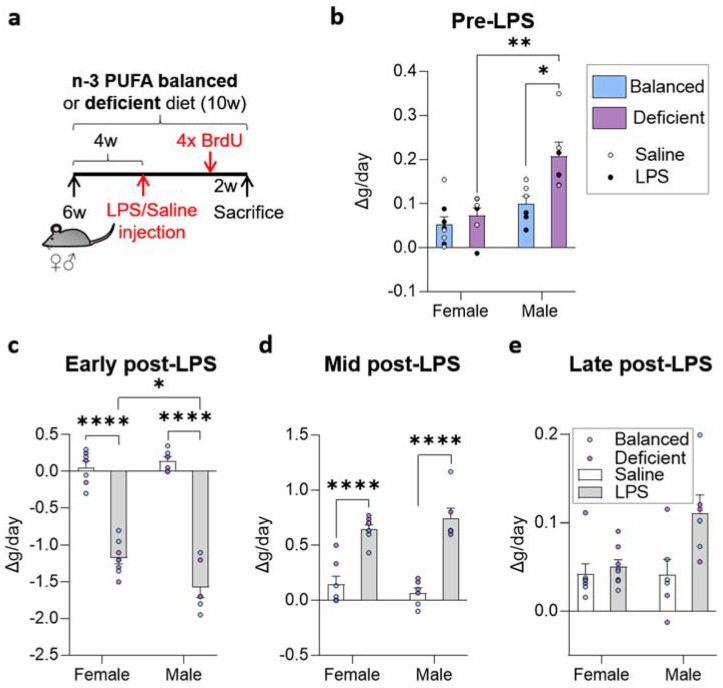
Experimental design and early effects of LPS treatment in mouse weight. (**a**) Experimental design. Mice were fed with the balanced or n-3 PUFA-deficient diets from 6 weeks of age, intraperitoneally injected with saline or lipopolysaccharide (LPS) to induce acute systemic inflammation at 10 weeks of age, and injected with BrdU four times 2 weeks before sacrifice to target newly generated cells. (**b**) Weight gain was higher in males than in female mice, and the n-3 PUFA-deficient diet increased weight gain before LPS administration (no interaction detected between diet and sex). (**c**) Weight loss induced by LPS treatment was not influenced by dietary n-6/n-3 PUFA ratio. Additionally, male mice lost more weight daily than females (significant interaction between LPS treatment and sex found). (**d**) After initial weight loss, LPS-treated mice recovered normal weight, as shown by the increase in weight change per day. (**e**) Most of the mice (treated with saline or LPS) slowly gained weight while aging. Please, note the different scales used for *y*-axes in b–e. * *p* < 0.05, ** *p* < 0.01, **** *p* < 0.0001 (statistics detailed in Appendix A).

**Figure 2 ijms-23-03399-f002:**
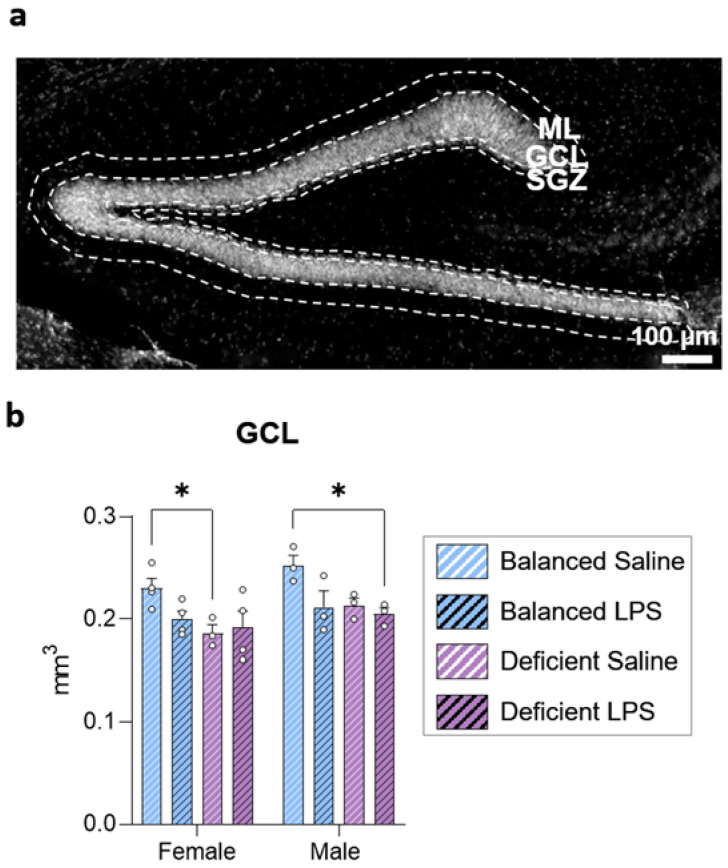
The n-3 PUFA deficient diet reduced the volume of the granule cell layer (GCL). (**a**) Epifluorescence image of the dentate gyrus showing the limits of the different layers (dotted lines): subgranular zone (SGZ), GCL, and molecular layer (ML). Images from each animal group are available in Appendix A. (**b**) We observed a significant reduction in the volume of the GCL of female mice treated with saline and fed with the n-3 PUFA-deficient diet compared to female mice fed with the balanced diet and the same treatment. Additionally, we detected a significant reduction in the volume of the GCL of male mice treated with LPS and fed with the n-3 PUFA-deficient diet compared to male mice fed with the balanced diet and treated with saline. * *p* < 0.05 (statistics detailed in Appendix A).

**Figure 3 ijms-23-03399-f003:**
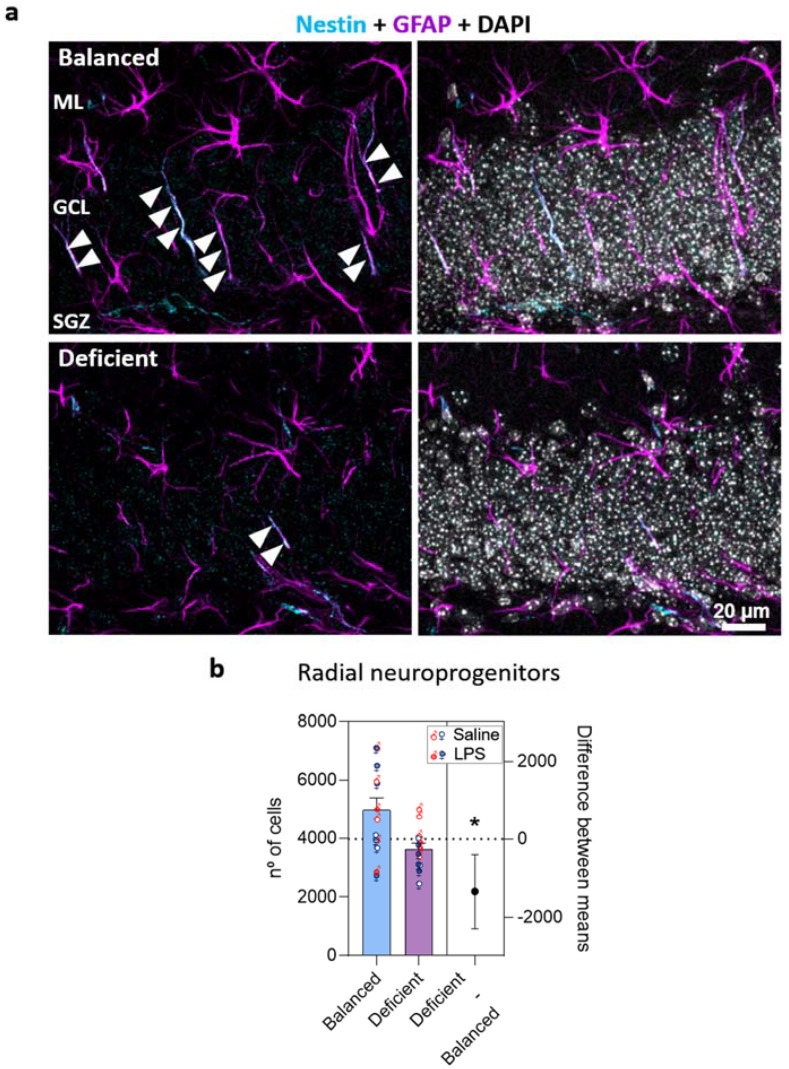
The n-3 PUFA-deficient diet reduced the number of radial progenitors in the dentate gyrus (DG). (**a**) Confocal images (maximum intensity projections, z = 13.8 µm) of the DG of the hippocampus showing radial neuroprogenitors identified by the expression of nestin (cyan) and GFAP (glial fibrillary acidic protein, magenta) and the presence of an apical process (arrowheads) crossing the granular cell layer of the DG. Additional images from each condition are available in Appendix A. (**b**) Estimation plots showing the diet effect size (difference between group means) and its associated 95% confidence interval (right axis). The number of radial neuroprogenitors was reduced in mice fed with the n-3 PUFA-deficient diet compared to mice fed with the balanced diet. No effects of LPS or sex were detected. * *p* < 0.05 (statistics detailed in Appendix A).

**Figure 4 ijms-23-03399-f004:**
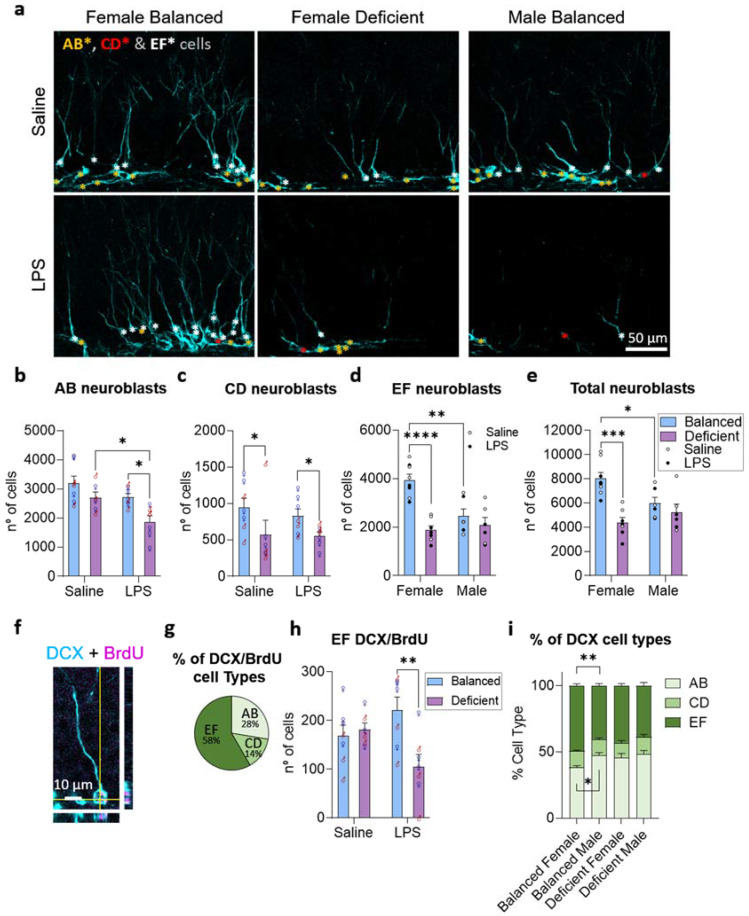
The n-3 PUFA-deficient diet reduced the number of neuroblasts in the dentate gyrus. (**a**) Confocal images (maximum intensity projections, *z* = 13.8 µm) of the dentate gyrus (DG) of the hippocampus showing neuroblasts expressing doublecortin (DCX, cyan) in female mice fed with the balanced or n-3 PUFA-deficient diets and treated with saline or LPS, and in male mice fed with the balanced diet and treated with saline or LPS. Different types of neuroblasts according to their morphology and maturation stage are identified with colored asterisks (early neuroblasts: AB cells in yellow, intermediate neuroblasts: CD cells in red, and young neurons: EF cells in white). (**b**) The n-3 PUFA-deficient diet and LPS treatment significantly reduced the number of early (AB) neuroblasts in the hippocampus (no effect of sex and no interaction between diet and LPS treatment detected). (**c**) Mice fed with the n-3 PUFA-deficient diet showed a reduced number of intermediate (CD) neuroblasts (no effect due to LPS treatment or sex detected). (**d**) Female mice fed with the n-3 PUFA balanced diet showed a higher number of young neurons (EF neuroblasts) than males fed with the same diet. The n-3 PUFA-deficient diet reduced the number of young neurons exclusively in female mice (interaction between diet and sex detected, no effect due to LPS treatment detected). (**e**) The total number of neuroblasts reflected the observations in EF neuroblasts. (**f**) Confocal image (maximum intensity projection and orthogonal views, *z* = 13.8 µm) showing a cell double-stained against DCX (cyan) and bromodeoxyuridine (BrdU, magenta). Additional images from different animal groups are available in Appendix A. (**g**) Pie chart showing the proportions of DCX- and BrdU-positive cells in each maturation stage (AB, CD, and EF). (**h**) Mice treated with LPS and fed with the n-3 PUFA-deficient diet showed a significant reduction in the number of double-stained DCX/BrdU cells compared to mice fed with the balanced diet (interaction between diet and LPS treatment detected, no effect due to sex detected). (**i**) Female mice fed with the n-3 PUFA-deficient diet showed a significant change in their proportion of AB and EF neuroblasts compared to male mice fed with the same diet (no interaction between diet and sex, and no effect of LPS treatment detected). * *p* < 0.05, ** *p* < 0.01, *** *p* < 0.001, **** *p* < 0.0001 (statistics detailed in Appendix A).

**Figure 5 ijms-23-03399-f005:**
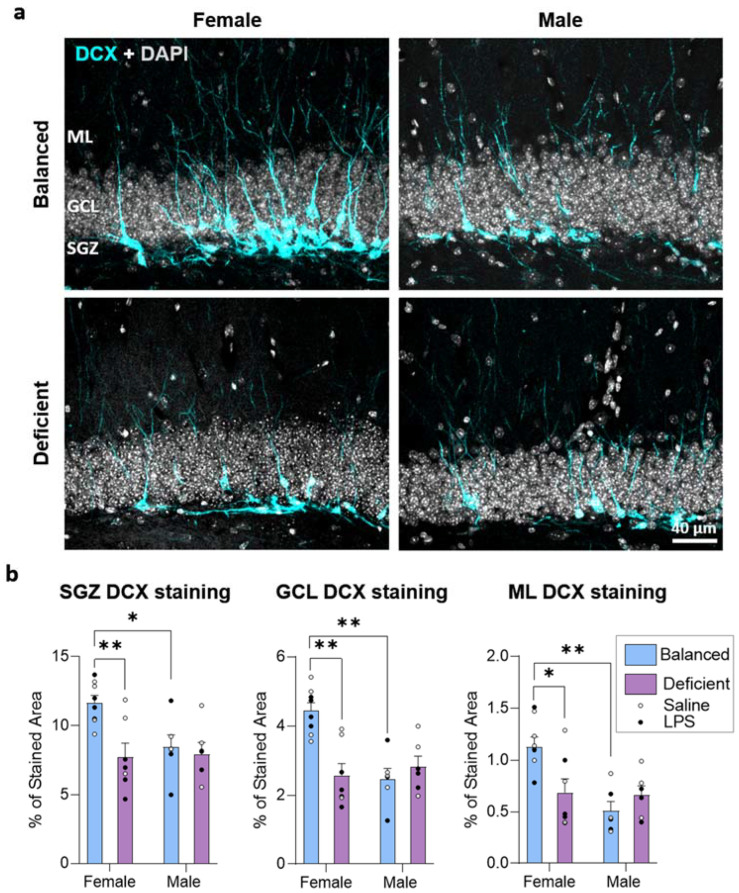
The n-3 PUFA-deficient diet reduced the proportion of area occupied by doublecortin (DCX) staining in the dentate gyrus of female mice. (**a**) Confocal images (sum of intensities projection, *z* = 13.8 µm) of the dentate gyrus (DG) of the hippocampus showing DCX (cyan) staining in the subgranular zone (SGZ), the granule cell layer (GCL), and the molecular layer (ML). (**b**) Female mice fed with the n-3 PUFA balanced diet showed a higher percentage of area stained for DCX than males fed with the same diet. The n-3 PUFA-deficient diet reduced the DCX staining exclusively in female mice (interaction between diet and sex detected, no effect of LPS detected). * *p* < 0.05, ** *p* < 0.01 (statistics detailed in Appendix A).

**Figure 6 ijms-23-03399-f006:**
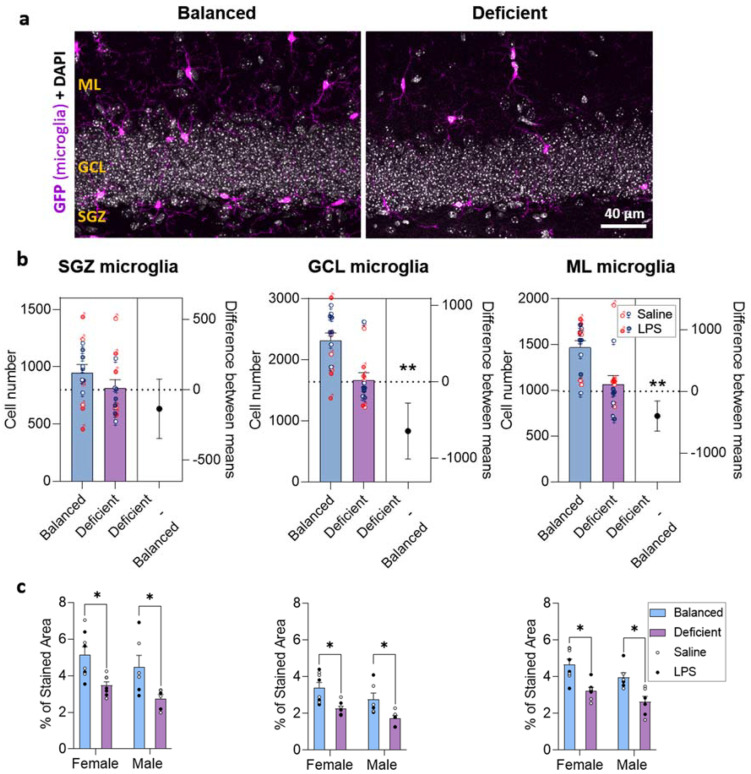
The n-3 PUFA deficient diet reduced microglia in the dentate gyrus (DG). (**a**) Confocal images (maximum intensity projections, *z* = 13.8 µm) of the DG of the hippocampus showing microglia expressing the green fluorescence protein (GFP, magenta) in the subgranular zone (SGZ), granular cell layer (GCL), and molecular layer (ML). Images from each animal group are available in Appendix A. (**b**) Estimation plots showing the diet effect size (difference between group means) and its associated 95% confidence interval (right axis). Mice fed with the n-3 PUFA-deficient diet showed fewer microglia in the GCL and ML than those fed with the balanced diet (no effects of LPS or sex detected). (**c**) Furthermore, the area occupied by microglia was reduced in the SGZ, GCL, and ML of mice fed with the n-3 PUFA-deficient diet compared to those fed with the balanced diet (interaction between diet and sex detected in the GCL and ML, no effect of LPS detected). * *p* < 0.05, ** *p* < 0.01 (statistics detailed in Appendix A).

**Figure 7 ijms-23-03399-f007:**
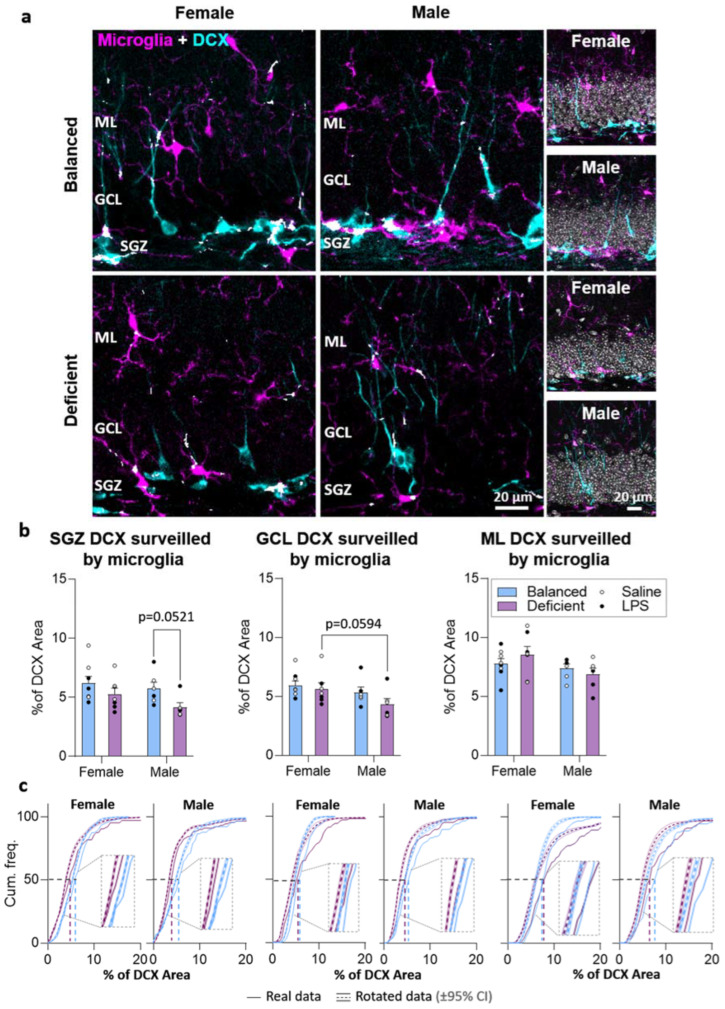
Reduced spatial proximity between microglia and neuroblasts in mice fed with the n-3 PUFA-deficient diet. (**a**) Confocal images (maximum intensity projections, *z* = 13.8 µm) of the dentate gyrus of the hippocampus showing neuroblasts expressing doublecortin (DCX, cyan), microglia expressing the green fluorescence protein (GFP, magenta), and the spatial colocalization of both stainings (white) in the subgranular zone (SGZ), granule cell layer (GCL), and molecular layer (ML) of female and male mice fed with the balanced or n-3 PUFA-deficient diets. Insets on the right also show DAPI staining in gray. Images from each animal group are available in Appendix A. (**b**) The proportion of area of neuroblasts (DCX expressing cells) surveilled by microglia (colocalizing with GFP) showed a trend toward a reduction in the SGZ of male mice fed with the n-3 PUFA-deficient diet compared to males fed with the balanced diet (no effect of LPS and no interactions detected, statistics detailed in Appendix A). In addition, in the GCL, male mice showed a trend toward a reduction in the area of neuroblasts surveilled by microglia compared to female mice (no effect of LPS and no interactions detected). (**c**). Our analysis of colocalization randomness showed that the actual proportion of the area of DCX neuroblasts colocalizing with microglia was higher than that obtained by artificially rotating microglia staining. Graphs show the cumulative frequency of real (straight lines) or rotated (colored dotted curves) images with a determined proportion of DCX area colocalizing with microglia. The black horizontal dotted lines show the 50% accumulative frequency, and the real percentage of colocalization at the 50% frequency is indicated by the vertical colored lines (note the difference between lines corresponding to different diets). Dotted rectangles show a magnification of the curves around the 50% cumulative frequency where the displacement of the real data to the right of rotated data is evident, indicating that real colocalization is higher than that expected by chance. Note the displacement to the right of data from mice fed with the balanced diet compared with data from mice fed with the n-3 PUFA-deficient diet, indicative of the significant effect of diet in microglial surveillance of neuroblasts (statistics detailed in Appendix A).

**Figure 8 ijms-23-03399-f008:**
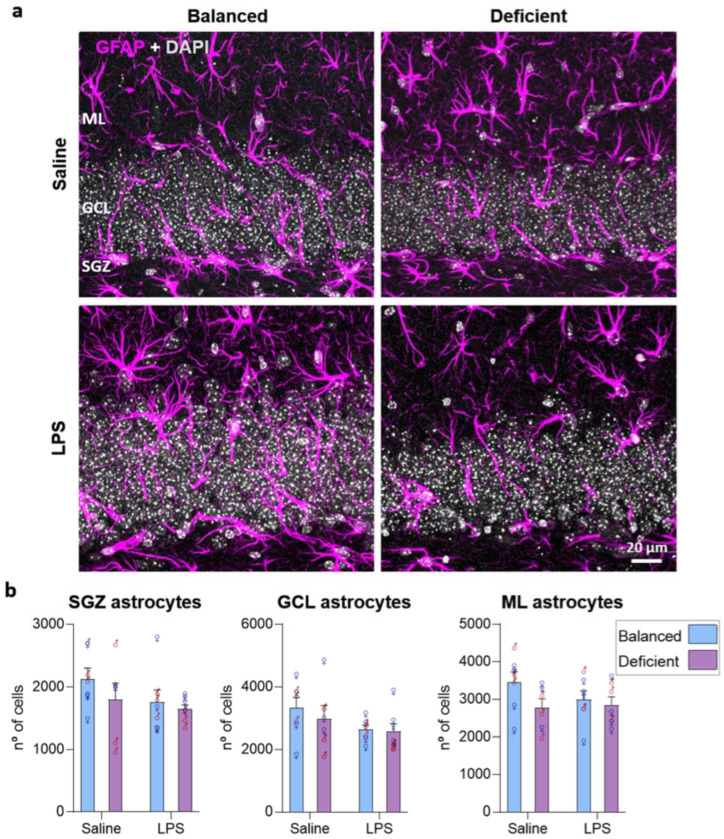
The n-3 PUFA-deficient diet had no relevant effects on astrocyte number. (**a**) Confocal images (maximum intensity projection, *z* = 13.8 µm) of the dentate gyrus of the hippocampus showing glial fibrillary acidic protein (GFAP, magenta) expression and DAPI staining (white) in the subgranular zone (SGZ), granular cell layer (GCL), and molecular layer (ML). (**b**) We did not detect consistent significant differences in the number of astrocytes between the different groups of mice analyzed (statistics detailed in Appendix A).

**Figure 9 ijms-23-03399-f009:**
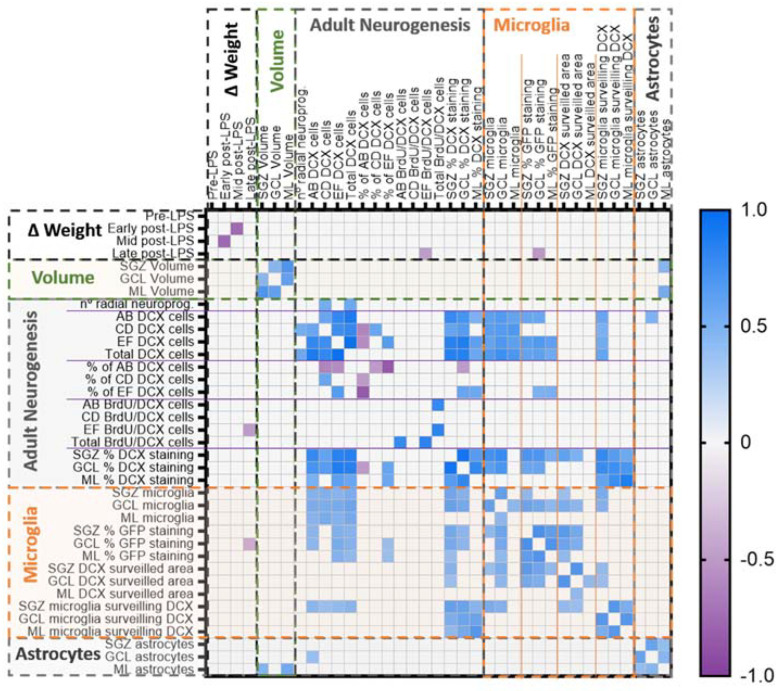
Correlation matrix analysis revealed multiple correlations between adult neurogenesis and microglia parameters. Dotted lines divide the different groups of parameters analyzed, and straight lines denote the subgroups of features related to adult neurogenesis. Only significant correlations (*p* < 0.01) are depicted as colored squares in which the color tone represents the Spearman’s *r* value (check scale bar in the right, single values detailed in Appendix A).

**Figure 10 ijms-23-03399-f010:**
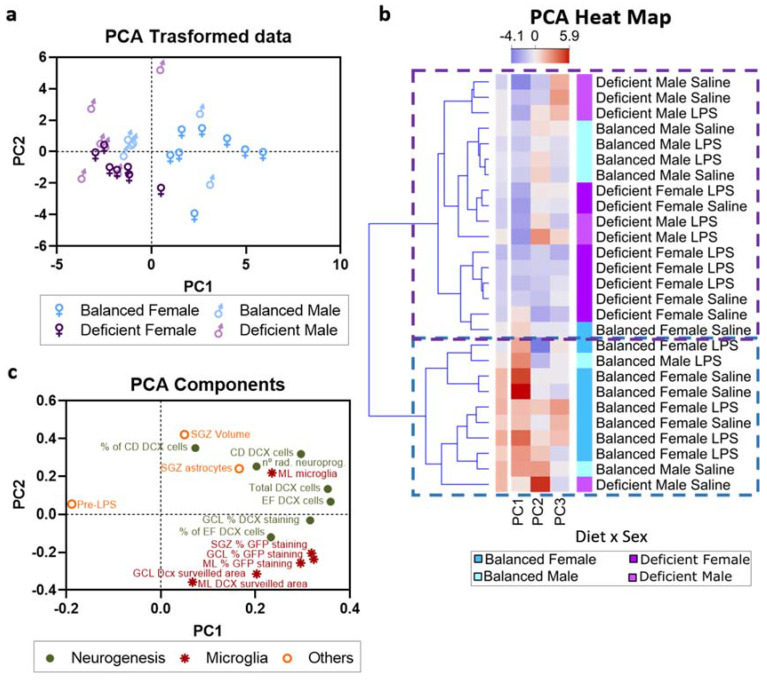
Principal component analysis (PCA) indicated a prominent effect of diet in female mice. (**a**) Principal component 1 (PC1) and principal component 2 (PC2) explained 62% of the variance of the selected data. The PCA-transformed data distribution showed relevant differences between female mice fed with the n-6/n-3 PUFA balanced diet and the other groups. (**b**) PCs 1, 2 and 3 were used to classify mice, and two main clusters were obtained (dotted lines). The heat map divided mice into two clusters (dotted lines). (**c**) PC1 was mainly influenced by adult neurogenesis and microglial-related parameters, while PC2 was mainly influenced by a mixture of features related to adult neurogenesis, microglia, and others (PCA values detailed in Appendix A).

**Table 1 ijms-23-03399-t001:** Composition of the diets (g/kg diet). 1: For a detailed composition of fatty acids, see Table 2. Composition (g/kg): sucrose, 110.7; CaCO_3_, 240; K_2_HPO_4_, 215; CaHPO_4_, 215; MgSO_4_·7H_2_O, 100; NaCl, 60; MgO, 40; FeSO_4_·7H_2_O, 8; ZnSO_4_·7H_2_O, 7; MnSO_4_·H_2_O. 2: CuSO_4_·5H_2_O, 1; Na_2_SiO_7_·3H_2_O, 0.5; AlK(SO_4_)_2_·12H_2_O, 0.2; K_2_CrO_4_, 0.15; NaF, 0.1; NiSO_4_·6H_2_O, 0.1; H_2_BO_3_, 0.1; CoSO_4_·7H_2_O, 0.05; KIO_3_, 0.04; (NH_4_)_6_Mo_7_O_24_·4H_2_O, 0.02; LiCl, 0.015; Na_2_SeO_3_, 0.015; NH_4_VO_3_, 0.01. 3: Composition (g/kg): sucrose, 549.45; retinyl acetate, 1; cholecalciferol, 0.25; dl-α-tocopheryl acetate, 20; phylloquinone, 0.1; thiamin HCl, 1; riboflavin, 1; nicotinic acid, 5; calcium pantothenate, 2.5; pyridoxine HCl, 1; biotin, 1; folic acid, 0.2; cyanocobalamin, 2.5; choline HCl, 200; dl-methionine, 200; *p*-aminobenzoic acid, 5; inositol, 10.

Ingredients	Amount
Casein	180
Cornstarch	460
Sucrose	230
Cellulose	20
Fat (1)	50
Mineral mix (2)	50
Vitamin mix (3)	10

**Table 2 ijms-23-03399-t002:** Fatty-acid composition of the dietary lipids (percentage of weight of total fatty acids). FAs: fatty acids; LA, linoleic acid; AA, arachidonic acid; PUFAs, polyunsaturated fatty acids; ALA, α-linoleic acid; ND, not detected by gas chromatography.

DIET	Ω6/Ω3 Balanced	Ω6/Ω3 Deficient
16:0	22.6	6.2
18:0	3.3	4.4
Other saturated FAs	1.8	1.6
Total saturated FAs	27.7	12.2
16:1 Ω7	0.2	0.1
18:1 Ω9	57.9	26.0
18:1 Ω7	1.5	0.9
Other monounsaturated FAs	0.4	0.2
Total monounsaturated FAs	60.0	27.2
18:2 Ω6 (LA)	10.6	60.5
20:4 Ω6 (AA)	ND	ND
Total Ω6 PUFAs	10.7	60.5
18:3 Ω3 (ALA)	1.6	0.1
18:4 Ω3	ND	ND
20:5 Ω3	ND	ND
22:5 Ω3	ND	ND
22:6 Ω3 (DHA)	ND	ND
Total Ω3 PUFAs	1.6	0.1
Total PUFAs	12.3	60.6
Ω6/Ω3 ratio	6.7	>500

## Data Availability

Data are contained within the article or Appendix A.

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
