# Peer review of "Susceptibility of Female Mice to the Dietary Omega-3/Omega-6 Fatty-Acid Ratio: Effects on Adult Hippocampal Neurogenesis and Glia"

_ijms, 2022, doi:10.3390/ijms23063399_

Round 1

Reviewer 1 Report

The paper submitted by Noelia Rodríguez-Iglesias and coauthors demonstrates the effect of a balanced omega-3/omega-6 fatty acid diet on neurogenesis and cell percentage in adult mice hippocampus. The authors performed a great job to evaluate the effects of unsaturated acids on the immunomodulatory potential and portion of different cell types in hippocampal layers. The authors also demonstrate that the observed effects of the diet are sex-dependent. The manuscript can be recommended for publication after minor revision. I have only some comments:

  1. Some diagrams are difficult to understand (3b, 4i, 6b etc). If it is possible, I would recommend revising them.
  2. In the discussion section, the numerous effects of the unsaturated acid are listed. Are any potential molecular mechanisms that may explain the observed effects? Please, discuss these mechanisms if they have been described in the literature. For instance, DHA, which is discussed in this section, may directly modulate the activity of ion channels and glutamate receptors, thus contributing to the regulation of synaptic plasticity and neurogenesis.

Reviewer 2 Report

It is an interesting work investigating the dietary effects on adult hippocampal neurogenesis. By feeding mice with a balanced or deficient n-3 PUFA diet, the author has found female mice fed by n-3 PUFA deficient diet has significantly reduced adult hippocampal neurogenesis, and decreased microglia number and surveilled volume. And the study has found n-3 PUFA deficient diet did not impact astrocyte in the adult brain. There are many previous reports in the area full of controversies. This work has shown some interesting data to support the study about adult neurogenesis. But some key experiment results shown here are incomplete or missing. The major concerns are as below:

  1. Please show the epifluorescence images of the dentate gyrus of each mouse from all treatment groups.
  2. In Figure 3, the author showed nestin and GFAP staining in DG. But there is no DAPI staining for counting the total cell number in the area or demonstrating the cell layer in the area exactly. How to confirm the dash lines in Fig 3a are correctly showing the granular cell layer of the DG? The author mentioned no effects of LPS or sex were detected in the figure legends. Where are those pictures? They should be shown here.
  3. In Figure 4, the author has described three types of DCX+ cells which are early neuroblasts (AB), intermediate neuroblasts (CD) and young neurons (EF). It is good to document different types of neuroblasts in the DG. But it can be not accepted to classify the cells by morphology only. The double staining with certain neuronal markers should be added here for verifying the classification at least, such PCNA or NeuN. The charts in Fig 4b are confusing. Please check to make sure all the labels including words on the X,Y axis are correct and matched to the results. For Fig 4F, how many BrdU and DCX positive cells have been found on the slide, only one? A typical DCX and BrdU staining picture of DG with clear view of the whole area under the confocal should be shown in the figure. It is required as  BrdU shown in Fig 4F is too weak to be taken as BrdU+ cell.
  4. The DAPI staining in Fig5 and Fig8 are too weak to be seen. The quality of the figures is not acceptable for publishing. Please replace them with prominent DAPI staining as those figures shown in the reference you have quoted.
  5. In Figure 6 and Figure 7, DAPI staining should be added together with antibody staining for showing the cell layer exactly but not only by a handmade dash line. Where are the staining pictures for showing the difference between saline group and LPS group? Please add them here.

Minor point:

The figure legends have contained much repeating sentences included in the results already. Please check the instruction for authors carefully at first and then to revise the figure legend part. Please check the abstract and conclusion part carefully to correct all typo and grammar mistakes.

Reviewer 3 Report

The authors did an excellent job on the influence of dietary n-6 / n-3 PUFA balance on adult neurogenesis and glia with gender differences.
Since the authors have already presented a lot of details in this paper, I will not suggest further analysis on the type of microglia. However, I strongly suggest that in this future work this should be done, it should certainly be checked what type of microglia is about, especially since the authors themselves claim that their results differ from those of other researchers in terms of upregulating microglia after n- 6 / n-3 PUFA ratio controlled diet. 

Round 2

Reviewer 2 Report

Thanks for the quick reply with all details. The revised manuscript is very impressive and more promising. It is a great work with a lot beautiful staining images. The new supplementary figures added into the manuscript strongly support the results and main conclusions. I'd be glad to suggest it for the acceptance as current form.